# Radio frequency interface quality assessment in 4G/5G: Effects of IQ imbalance, phase noise, and nonlinearities on error vector magnitude

Ilya Pyatin[1], Juliy Boiko[2], Viacheslav Kovtun[3]*, Oksana Kovtun[4]

1 Department of Computer Engineering, Khmelnytskyi Polytechnic Professional College by Lviv Polytechnic National University, Khmelnytskyi, Ukraine, 2 Department of Telecommunications, Media and Intelligent Technologies, Khmelnytskyi National University, Khmelnytskyi, Ukraine, 3 Computer Control Systems Department, Vinnytsia National Technical University, Vinnytsia, Ukraine, 4 Department of the Theory and Practice of Translation, Faculty of Foreign Languages, Vasyl' Stus Donetsk National University, Vinnytsia, Ukraine

* kovtun_v_v@vntu.edu.ua

## Abstract

Modern 4G/5G technologies aim to enhance data speeds, improve communication quality, and enable innovative services such as IoT and augmented reality. However, their efficiency depends on minimizing distortions in the radio frequency (RF) interface, evaluated through Error Vector Magnitude (EVM). Increased EVM leads to packet losses and reduced throughput, making its reduction essential for stable and high-quality networks. This study investigates the impact of RF interface imperfections on EVM in 4G/5G systems. The analysis was conducted using Simulink models of digital communication transmitters and receivers, incorporating in-phase and quadrature (IQ) imbalance, phase noise, power amplifier (PA) nonlinearity, channel noise, and signal-coding scheme characteristics. The QM78207 chipset, integrating key RF components, was used as an example to reflect the complexity and quality requirements of modern RF interfaces. The results show that the maximum allowable EVM for 64-QAM is 8% (-22 dB). Variations in IQ amplitude imbalance (0–3 dB) increased EVM from -32 dB to -15 dB, while IQ phase imbalance (0°–15°) caused an increase from -32 dB to -17 dB, both for SNR = 50 dB. These findings are valuable for optimizing RF interface designs in 4G/5G systems, ensuring enhanced communication quality and supporting the growing demands for advanced services.

## 1. Introduction

### 1.1. Formulating research issues

RF Front End plays a critical role in ensuring high-quality communication and data transmission in modern mobile telecommunication systems [1]. Situated between the antenna switch and the mixer, it facilitates signal matching with the antenna and

**Data availability statement:** All relevant data are within the paper.

**Funding:** The author(s) received no specific funding for this work.

**Competing interests:** The authors have declared that no competing interests exist.

converts signals into a format suitable for further processing or radiation into free space [2]. To support various communication standards (4G, 5G, Wi-Fi, GPS) [3–6] utilized by mobile devices, the RF Front End provides distinct signal transmission and conversion paths for each standard [7,8]. However, existing studies lack a comprehensive analysis of the relationship between signal quality and the signal-to-noise ratio (SNR). One of the critical metrics for assessing signal quality is the EVM. The objective of this study is to investigate the impact of RF Front End imperfections on EVM in LTE and 5G systems. The EVM quantifies the difference between ideal and measured symbols after equalization. The EVM result is defined as the root mean square (RMS) of the error vector power relative to the average reference power, expressed as a percentage. To enhance the interpretability of results, EVM values are presented on graphs in decibels (dB).

The impact of the power amplifier's nonlinearity on the reliability of the received digital signal was determined using the EVM parameter [9]. We used this fundamental metric in our work to conduct experimental studies. The expression for the EVM error vector is defined as the difference between the actual $v(\tilde{I}, \tilde{Q})$ and the expected $w(I, Q)$ symbol of the modulated signal on the state diagram in the orthogonal $I/Q$ coordinate system:

$$EVM = w(I, Q) - v(\tilde{I}, \tilde{Q}) \qquad (1)$$

We present the expression in the following form:

$$EVM = \frac{1}{|w_{max}|} \sqrt{\frac{1}{N} \sum_{j=1}^{N} \left(I_j - \tilde{I}_j\right)^2 + \left(Q_j - \tilde{Q}_j\right)^2} \qquad (2)$$

where $|w_{max}| = \sqrt{I_{max}^2 + Q_{max}^2}$

Modern communication technologies, including LTE and 5G NR, as emphasized earlier in the text, aim to achieve high data transmission rates, ensure stable communication quality, and enable innovative services such as the Internet of Things (IoT) and augmented reality (AR). However, the efficiency of these technologies heavily depends on minimizing distortions in the RF interface, which are evaluated using the EVM metric. The primary challenge is that an increase in EVM leads to packet loss, reduced network throughput, and degraded communication quality.

In the context of high-speed standards such as LTE Advanced Pro, 5G NR, and advanced technologies like 256-massive MIMO and 64-QAM/256-QAM, sensitivity to distortions becomes significantly higher. Therefore, this study aims to investigate the impact of RF interface hardware imperfections—such as IQ imbalance, phase noise, PA nonlinearity, and channel noise—on the EVM metric. Addressing this scientific challenge is critical, as failure to mitigate these limitations hinders the ability to provide stable network per-formance capable of meeting the growing demand for high-quality services. To address the outlined problem, this paper will model the influence of RF interface hardware imperfections on signal modulation quality in 4G/5G

systems. The focus will be on identifying critical parameter levels (*IQ* imbalance, phase noise, PA nonlinearity) that directly affect the EVM metric in high-speed systems, serving as baseline references for comparison.

Based on the results presented in this paper, we will develop recommendations for optimizing RF interface design to minimize signal distortions. These recommendations will contribute to improving communication quality in modern telecommunication systems utilizing 4G/5G standards.

## 1.2. Literature review

We will review advanced research published in renowned high-ranking journals on the topic addressed in the presented article. In [1], the focus is on the analysis of RF signal distortion models and compensation algorithms in single-carrier and multicarrier systems, such as Orthogonal Frequency Division Multiplexing (OFDM). The authors also examine the impact of RF distortions in advanced technologies, including MIMO, massive MIMO, full-duplex communication, and millimeter waves. The value of their work lies in the comprehensive review of evaluation and compensation algorithms, as well as in identifying directions for future research, including the use of artificial intelligence for developing compensation algorithms. However, their study is largely concentrated on generalized models of impact and compensation algorithms. It lacks a detailed analysis of specific distortion parameters, such as IQ imbalance, phase noise, and power amplifier nonlinearity, within the context of specific 4G/5G systems, as well as an assessment of how these distortions affect EVM, which is a critical metric for modern networks.

The article [2] focuses on the development of novel architectural components for 5G RF front-end modules. It proposes designs and measurements aimed at reducing RF noise, improving module efficiency, and enabling simultaneous operation of multiple transmitters. While the article emphasizes engineering solutions for reducing RF noise and ensuring efficient operation of RF modules, it does not analyze specific RF interface distortions or their impact on EVM performance. Moreover, practical recommendations for optimizing RF design to improve communication quality are not addressed.

The study [3] provides an indepth examination of OFDM, its performance in multipath channels, and its behavior under various distortions. The article also reviews key alternative waveform candidates that offer better time and frequency localization compared to OFDM, leveraging filtering or windowing techniques. The paper highlights the trade-offs of these approaches in the context of designing waveforms for next-generation wireless communication technologies. However, it lacks a detailed analysis of specific RF distortion parameters, such as IQ imbalance or phase noise, which critically impact communication quality in 4G/5G networks, particularly when using OFDM modulation. This omission indicates a gap in addressing practical optimization of RF design for 4G/5G systems.

The work [10] focuses on developing a DPD/*IQ* method that compensates for in-phase/quadrature (I/Q) imbalance and PA nonlinearity in RF systems. The method is tailored for use in base stations employing low-cost software-defined radio (SDR). The proposed techniques provide an effective solution for mitigating I/Q imbalance and PA nonlinearity, with an emphasis on implementation in SDR-based systems. However, the study does not analyze or investigate various RF interface distortions or their impact on EVM in modern 4G/5G networks.

In [11], the authors analyze the impact of major distortions in the RF front-end that affect the performance of modern wireless communication systems. The focus is on nonlinear PA distortions and amplitude-phase imbalances in the transmitter mixer. However, the study provides only an overview of distortions without presenting a quantitative analysis of their impact on key metrics such as EVM, which are critical for contemporary 4G/5G networks. Furthermore, the work lacks a quantitative assessment of how these distortions affect communication quality. The omission of EVM as a key metric and the absence of practical approaches for optimizing RF systems further limit the scope of the study.

The article [12] examines the combined impact of IQ imbalance and carrier phase/frequency offset. The authors use a system-level (SL) model to evaluate these effects, incorporating measurements under various local oscillator (LO) configurations (coherent and non-coherent) and analyzing dependencies on bandwidth and frequency. The study evaluates EVM, considering recording time, frame evaluation periods, and averaging based on carrier frequency offset (CFO).

However, it does not address the assessment of phase noise, PA nonlinearity, channel noise, or signal coding characteristics. While the authors conduct EVM measurements, their analysis is not linked to real-world network characteristics, such as throughput or packet loss, which would enable direct optimization of RF interfaces for modern 4G/5G networks.

The research presented in [13] explores spectral analysis mechanisms enabling secondary users (SUs) to utilize the spectrum efficiently without causing interference to primary users (PUs). The authors propose a methodology that accounts not only for IQ imbalance but also for aliasing effects, which is critical for receivers with limited performance. However, the study does not address the impact of phase noise, amplifier nonlinearity, or channel noise. The article focuses on narrow aspects of cognitive radio network development but neglects practical considerations for optimizing RF interfaces for modern 4G/5G networks, including a broader spectrum of signal distortions.

In [14], the authors propose modules for evaluating and compensating for frequency-dependent (FD) IQ imbalance and group delay in millimeter-wave (mmWave) systems. They demonstrate that employing digital recursive filters to mitigate FD IQ imbalance and group delay improves parameters such as the image rejection ratio (IRR), increasing it from 21.97 dB to 47.54 dB. The proposed techniques support a single-carrier (SC) mode with a throughput of 15 Gbps, which is critical for 5G and 6G networks. However, other imperfections, such as phase noise, power amplifier nonlinearity, channel noise, and the impact of signal-coding schemes, are not addressed in the study. The authors focus exclusively on IQ imbalance and group delay.

In [15], the authors present an innovative method for compensating IQ imbalance in sensing systems with a simplified architecture and reduced computational requirements. They propose a novel controlled feedback structure for phase demodulation based on a zero-error closed-loop design. The study addresses the reduction of distortions caused by IQ imbalance and the optimization of noise levels without requiring complex IQ evaluation and compensation. However, the practical functionality of real RF interfaces in 4G/5G systems is not considered. Additionally, the paper omits critical issues such as phase noise, power amplifier nonlinearity, and channel noise analysis.

The presented work [16] describes a method for characterizing signal distortion caused by amplifiers under modulated operating conditions. The method is based on decomposing the output signal into a part that is linearly correlated with the input signal, and another part that represents nonlinear distortion. The decomposition is carried out by calculating the statistical cross-correlation between the spectra of the input and output signals. The measurements are performed using a vector network analyzer. However, this paper does not address specific methods for compensation or distortion estimation in real-world systems, considering the impact of RF distortions on 64-QAM modulation through detailed EVM assessment. Additionally, the integration of methods for compensating IQ imbalance is insufficiently described, which is a critical aspect for the accurate analysis of system performance in applications with high signal fidelity requirements.

In the paper [17], a frequency-domain method for measuring EVM in vector signal generators (VSG) is proposed, which allows for accurate results without the use of broadband receivers or waveform reconstruction, while accounting for the IQ imbalance effect. However, the impact of RF distortions on 64-QAM modulation, which is crucial for precise measurements in modern communication systems, is not addressed. Additionally, methods for compensating IQ imbalances, which are essential for high-precision measurements, are not mentioned. The paper also lacks a detailed analysis of distortions and the integration of compensation methods that would enable more accurate assessment of system performance.

Based on a comprehensive analysis of the works and studies of leading authors, we have identified several key shortcomings in previous research that our article aims to address. Most existing studies lack a holistic analysis of the impact of distortion parameters on the EVM in 4G/5G systems. Prior research primarily focuses on general distortion models or isolated aspects, such as IQ imbalance or amplifier nonlinearities, without considering the combined effects of these distortions alongside phase noise, channel noise, and signal coding characteristics. Furthermore, we observed an insufficient connection between distortion analysis and the practical requirements of real-world systems. Many works fail to provide quantitative assessments of distortion impacts on key performance indicators, such as EVM, throughput, or packet loss, limiting their practical relevance for optimizing RF interfaces. Additionally, limited attention has been given to modern

hardware platforms, with most studies focusing on theoretical models and overlooking inte-gration opportunities with advanced chipsets, such as the QM78207, which could enhance analysis accuracy and address contemporary RF quality requirements. Finally, there is a notable tendency in previous works to overlook certain critical types of distortions. For instance, phase noise, PA nonlinearities, and channel noise are seldom comprehensively studied, leaving gaps in under-standing their impact on communication quality. This article addresses these critical gaps by providing detailed modeling and analysis of distortions using Simulink, delivering practical value for RF design optimization in 4G/5G systems and enhancing communication quality.

Thus, after analyzing the works of renowned authors, we emphasize that the proposed study aims to address and complement relevant challenges in the field. The presented article examines overlooked aspects that represent gaps in previous research, focusing on a detailed analysis of the impact of RF distortions on EVM in 4G/5G systems. The study employs a Simulink-based approach to simulate transmitters and receivers, accounting for IQ imbalance, phase noise, power amplifier nonlinearity, channel noise, and signal-coding scheme characteristics. This approach enabled a quantita-tive assessment of the impact of each type of distortion.

Additionally, the study incorporates a specific hardware example — the QM78207 chipset, which integrates key RF components. This allowed for the consideration of modern requirements for RF interface quality and complexity. There-fore, the approach presented in the article, unlike existing studies, provides practical value for optimizing RF design in 4G/5G networks, contributing to improved communication quality and the support of innovative and cutting-edge services.

This study highlights how imperfections such as IQ imbalance, phase noise, and PA nonlinearities influence EVM, con-tributing to the optimization of RF Front End designs in advanced telecommunication systems. The findings are valuable for improving communication quality and ensuring compliance with evolving 4G/5G standards.

The structure of the article follows a logical and consistent format: Introduction - this section highlights the key advance-ments in the subject area of the research, identifies existing challenges, and addresses the methodological and prac-tical gaps that this study aims to resolve; Materials and Methods - here, we present the methodologies and theoretical foundations employed in the study, including simulation models. This section outlines the mathematical framework and emphasizes the influence of key parameters on the results; Results - this section provides the core experimental findings of the research, showcasing the outcomes obtained through the proposed methods; Discussion - dedicated to analyzing the main conceptual findings, this section elaborates on their significance, specific characteristics, and potential practical applications in 4G/5G technologies; Conclusion - summarizes and systematizes the principal results of the study, offering a clear overview of its contributions; References - contains a comprehensive list of the literature sources used to ensure the quality and credibility of the article. This structured approach, in our opinion, ensures clarity and facilitates a compre-hensive understanding of the research and its implications.

## 2. Materials and methods

### 2.1. Methodologies for the analysis of distortion and intersection points in narrowband circuits for evaluating the linearity of power amplifiers and intermodulation products

Modern mobile phones feature a transmitter and receiver housed within a single enclosure. To shift the signal spectrum from the baseband to the carrier frequency, a double frequency conversion process is employed. The RF interface is essential for generating the signal to be transmitted to the antenna, or for the initial amplification and signal shaping for the receiver. In the receiver, the analog processing includes filters, a low-noise amplifier (LNA), and down-conversion mixers, which are necessary for processing the signals received by the antenna into forms suitable for processing in the baseband processor [18].

For the transmitter, the baseband analog signals must be up-converted to the radio frequency range using mixers, and then amplified by a PA before being transmitted by the antenna. Fig 1 illustrates the block diagram of a modern mobile phone.

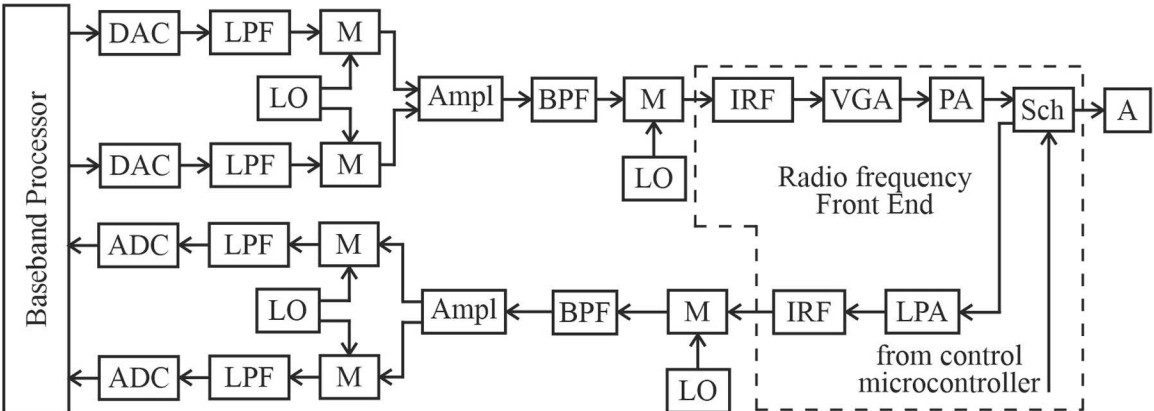

**Fig 1. Block diagram of a mobile phone.** DAC is the Digital to Analog Converter; LPF is the Low Pass Filter; M is the Mixer; LO is the Local Oscillator; ADC is the Analog to Digital Converter; Ampl is the Amplifier; BPF is the Band Pass Filter; IRF is the Image Rejection Filter; LPA is the Low Power Amplifier; VGA is the Variable Gain Amplifier; PA is the Power Amplifier; Sch is the Switch or Duplexer; A is the Antenna.

To improve the output power level, a VGA is used for pre-amplification of the RF signals before the PA (Fig 1).

The main objectives of LTE and 5G are to achieve low latency and high data transfer rates [19,20]. The capacity of a wireless system is determined by Shannon's capacity formula:

$$C = W \log_2 \left( 1 + S/N \right) \tag{3}$$

where $C$ is the channel capacity; $S$ is the average received signal power; and $N$ is the average noise power.

As demonstrated through the analysis of works by prominent authors and the formulation of the research problem, the quality assessment of RF interfaces in 4G and 5G networks has become a critical aspect of ensuring robust data transmission while maintaining the required level of Quality of Service (QoS). Specifically, issues such as in-phase and quadrature (IQ) imbalance, phase noise, and system nonlinearities significantly affect the EVM, a key parameter for evaluating the performance of digital RF transmitters and modulators.

Advanced research in this field focuses on the development of precise methodologies for EVM evaluation and the analysis of how nonlinear parameters and noise impact signal transmission while minimizing degradation in QoS metrics. The most notable findings highlight the importance of sequential evaluation of IQ imbalance and phase noise in determining the quality of transmitted signals [10–13].

Innovations in the design of modulation devices, particularly IQ modulators, have introduced comprehensive methods for measuring and analyzing spectral components arising from the formation of complex envelopes. Previous studies have shown a significant reduction in EVM through the optimization of IQ modulation system parameters.

The methodology proposed in this in the following sections systematically explains how IQ imbalance, phase noise, and nonlinear distortions influence EVM. The analysis is grounded in mathematical models and practical experiments, with a primary focus on measuring the spectral density of signals and their correlation with the parameters of transmission devices. This provides a comprehensive study framework, bridging theoretical foundations and practical implementation.

In narrowband systems, distortions are typically measured by applying two sinusoids with frequencies $f_1$ and $f_2$ that lie within the system's bandwidth. The harmonics of these two frequencies are located outside the system's bandwidth, while the distortion products at frequencies $2 f_1 - f_2$, $2 f_2 - f_1$, $3 f_1 - 2 f_2$, $3 f_2 - 2 f_1$ fall within the system's bandwidth.

The intersection point of the $n$-th order, $IP_n$, is defined in terms of the power levels of the fundamental tones and the n-th order distortion products, extrapolated based on their values at low signal levels.

$$IP_3 = P_{in} + \frac{\Delta P}{2}$$

<div align="right">(4)</div>

where $IP_3$ is the $n$-th order intercept point (dBm); $P_{in}$ is the input power of the fundamental frequency in dBm; $\Delta P$ is the difference between the desired output signal and the unwanted $n$-th order distortion product in dB; $P$ is the represents the input power.

Fig 2 illustrates the Simulink model of the RF Transmitter block, which consists of three main components: a baseband signal generator, an RF transmitter for converting the signal to the carrier frequency, and a receiver with a demodulator for EVM calculation [21].

Fig 3 shows the spectral density of the signal at the output of the quadrature modulator (IQ Modulator) under the influence of two input signals with frequencies of 10 MHz and 15 MHz. The sampling frequency is 100 MHz, and the local oscillator frequency is 2 GHz.

The PA can operate as a linear device only within a limited input power range; as the input power increases, the signal becomes distorted. The linear and nonlinear operating regions of the PA can be defined relative to its 1 dB compression point, where the PA's output power decreases by 1 dB, as shown in Fig 4 [22]. The third-order intercept point ($IP_3$) is the intersection of the asymptotes of the third-order intermodulation product and the fundamental signal. $iIP_3$ represents the input power, and $oIP_3$ represents the output power corresponding to $IP_3$.

When calculating $IP_3$, only the first two odd-order terms (4) should be considered.

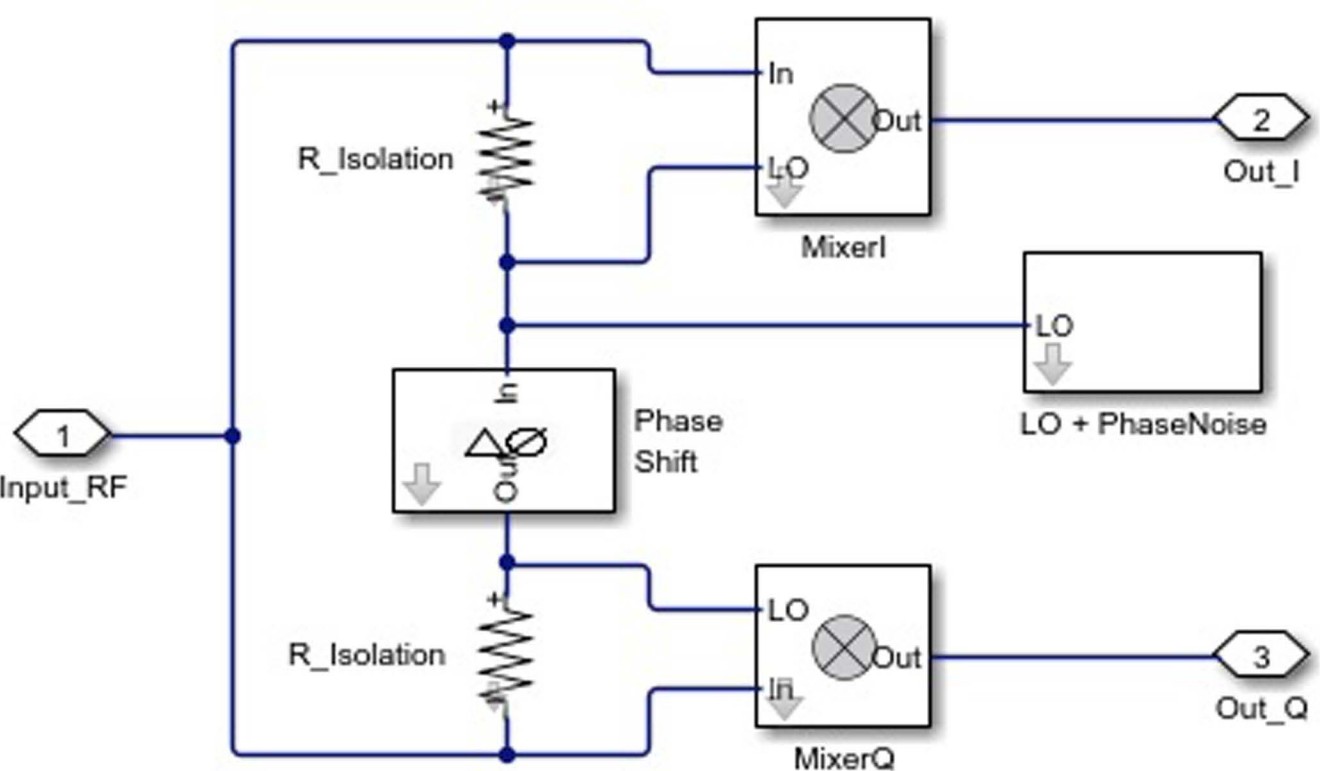

**Fig 2. Simulink model of the RF Demodulator block.**

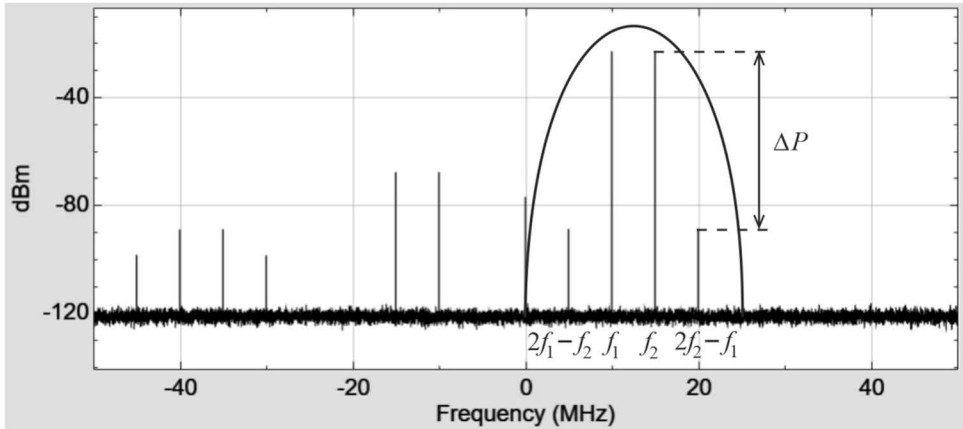

**Fig 3. The spectral density of the signal at the output of the IQ Modulator.**

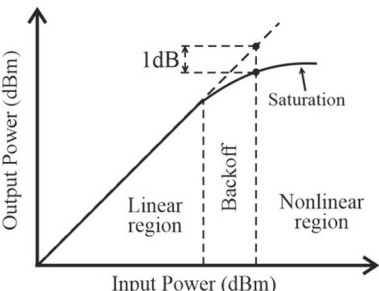

**Fig 4. The relationship between the input power and output power of the PA [22].**

$$x = au + cu^3 \tag{5}$$

We applied a two-tone test signal.

$$u(t) = \alpha \cos \omega_1 t + \beta \cos \omega_2 t \tag{6}$$

We substitute the input signal into the approximating polynomial and obtain:

$$x(t) = a\left(\alpha \cos \omega_1 t + \beta \cos \omega_2 t\right) + c(\alpha \cos \omega_1 t + \beta \cos \omega_2 t)^3 \tag{7}$$

The following trigonometric identities will be used to separate the equation into terms associated with individual frequencies.

After combining like terms, we obtain:

$$x(t) = \left(a\alpha + \frac{3c\alpha^3}{4} + \frac{3c\alpha\beta^2}{2}\right)\cos \omega_1 t + \frac{c\alpha^3}{4}\cos 3\omega_1 t +$$

$$\left(a\beta + \frac{3c\alpha^2\beta}{2} + \frac{3c\beta^3}{4}\right)\cos \omega_2 t + \frac{c\beta^3}{4}\cos 3\omega_2 t +$$

$$\frac{3c\alpha^2\beta}{4}\left(\cos(2\omega_1 t + \omega_2 t) + \cos(2\omega_1 t - \omega_2 t)\right) +$$

$$\frac{3c\alpha\beta^2}{4}\left(\cos(2\omega_2 t + \omega_1 t) + \cos(2\omega_2 t - \omega_1 t)\right) \tag{8}$$

Let $p_{o1} = (a\alpha)^2$ be the output power at $\omega_1$, $p_{o2} = (\alpha\beta)^2$ the output power at $\omega_2$, and $p_{o12} = (3c\alpha^2\beta/4)^2$ the output power at $|2\omega_1 \pm \omega_2|$. Now, apply the aforementioned methodology with $n=3$ to calculate $oIP_3$.

Then, to calculate $iIP_3$, using a similar methodology with $p_{i1} = \alpha^2$ as the input power at $\omega_1$, we obtain the expression for $oIP_3$:

$$oIP_3 = dB_{20}\left(\sqrt{\left|\frac{4a}{3c}\right|}\right) \tag{9}$$

As before, we note that $iIP_3$ and $oIP_3$ are completely independent of $\alpha$ and $\beta$, the amplitudes of the two test tones. Therefore, there is no requirement for the two input tones to have equal amplitude.

Now, let's consider that the Inport block (see Fig 2) allows specifying the complex envelopes of the input signals and importing them as RF signals. The complex signals at the input of the IQ modulator, $I_{IF}(t) + jQ_{IF}(t)$, have an intermediate frequency $f_{IF}$, and at the output, they have the carrier frequency ($f_c$), which we will represent by the following expression:

$$out = \mathrm{Re}\left(\sum_{k=1}^{N}(I_c(t) + jQ_c(t)) \cdot e^{j2\pi f_c t}\right) \tag{10}$$

## 2.2. Methodology for the analysis of simulink models of a radio frequency receiver and demodulator under conditions of IQ-imbalance, instability of quadrature components, phase noise, and carrier phase offset

In fact, we have depicted this process in Fig 3. Next, the transformation of the complex envelopes of the signal takes place: filtering, pre-amplification using the VGA, and final power amplification.

Below, we propose a model of a RF receiver, as presented in Fig 5, which serves as a comprehensive framework for analyzing the impact of phase noise, IQ imbalance, and nonlinearity on the EVM in 4G/5G networks. This methodology aligns with modern trends in wireless communication systems, emphasizing high-precision signal processing and reliable quality metrics.

The developed RF demodulator model incorporates key components such as a LO with phase noise simulation, phase shift generation for the in-phase (I) and quadrature (Q) components, as well as mixers. These elements enable the simulation of real-world distortions commonly encountered in RF systems. Phase noise from the LO directly affects signal coherence, causing distortions in the demodulated signal. The modeling of phase noise within the proposed methodology allows for an accurate assessment of its impact on EVM.

We also detail IQ imbalance, which typically arises from discrepancies in gain or phase between the I and Q channels. This imbalance degrades signal quality by introducing errors into the constellation points, especially for higher-order modulation schemes like M-QAM. Furthermore, we emphasize the critical effect of mixer nonlinearity. The RF-to-LO isolation parameter determines the effectiveness of suppressing undesired frequency components, ensuring the model's fidelity to real-world operating conditions.

Key features of the proposed methodology include the use of a matched filter to maximize suppression of intersymbol interference (ISI) under Nyquist conditions. The obtained in-phase and quadrature components provide high precision in

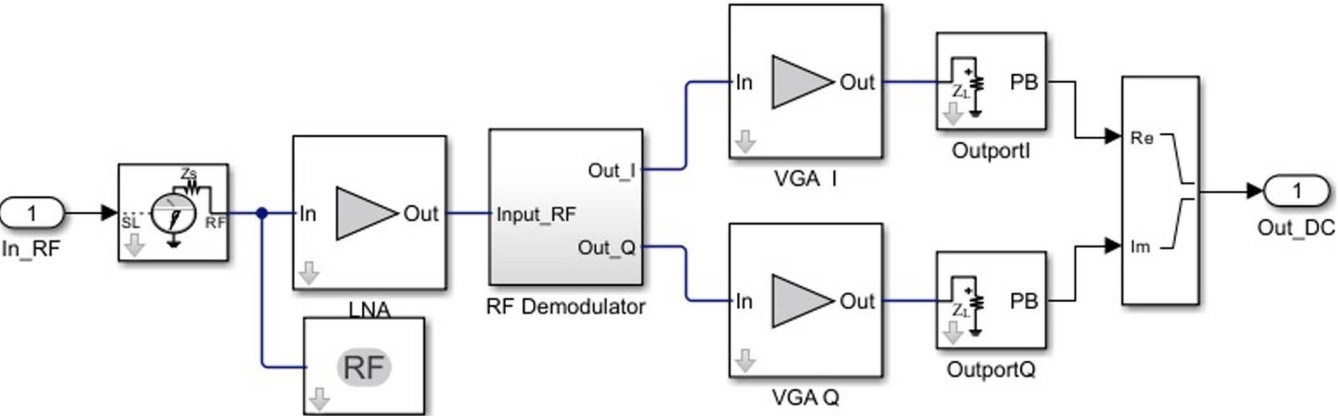

**Fig 5. Simulink model of the Decode Subframe block.**

symbol detection. Moreover, incorporating autocorrelation functions into the mathematical model enables detailed characterization of the influence of pulse shaping filters, which significantly affect EVM. This will be demonstrated in section 3 of this article, where we present the results of experimental studies.

The proposed Simulink models for the RF Transmitter (Fig 5) and RF Demodula-tor (Fig 2) enable precise analysis of practical distortions in a controlled environment. These models facilitate system parameter optimization in a practical context to achieve high performance.

In the context of methodological development and research objectives, the proposed approach addresses a critical issue: evaluating EVM under the combined influence of phase noise, IQ imbalance, and non-linearity. The RF receiver model and corresponding mathematical framework provide a comprehensive methodology for assessing the impact of RF component distortions on signal quality. The integration of Simulink simulations and theoretical analysis creates a robust toolkit for RF system optimization.

Most existing studies (as established in the Introduction through a thorough review of advanced works in the field [1–14]) analyze these distortions separately, whereas the model proposed in this paper integrates their cumulative effect. It is important to highlight that the presented research is particularly relevant in the context of 5G technology implementation, where high-order modulation schemes and wide bandwidths demand stringent EVM control.

Thus, the proposed methodology contributes to the development of robust communication systems that meet the requirements of modern wireless networks. The research results presented in section 3 of this article below highlight the importance of continuously improving the methods for assessing the quality of RF interfaces to ensure the reliability and efficiency of next-generation communication systems.

Fig 5 presents the model of the radio frequency receiver.

Note that the RF demodulator includes the following components: LO and PN model; phase shift for the I and Q component generator; mixers.

Below, we provide a detailed description of the subsystem blocks used in the model (Fig 5), as well as the demodulator subsystem presented in Fig 2.

Initialization of block parameters is performed by clicking on the block icon in the Simulink environment. This interactive approach enables on-the-fly configuration of parameters during experimentation. Additionally, clicking on the arrow icon within a Simulink block structure provides access to the internal architecture of the respective subsystem. In the following, we elaborate on the functionality and structure of the main blocks utilized in the proposed models.

The Inport block imports the external RF signal into the Simulink environment, serving as the input source for subsequent processing stages. The LNA amplifies the signal to a level suitable for further stages while minimizing added noise. The RF Demodulator performs frequency downconversion, shifting the RF signal to an intermediate frequency (IF), thus preparing it for digital signal processing and data extraction. We highlight that the demodulator subsystem (Fig 2) includes a LO that allows the simulation of phase noise, contributing to quadrature instability, as well as a block for applying phase shifts to the I/Q channels.

The VGA dynamically adjusts the gain to optimize signal amplitude for downstream processing (e.g., analog-to-digital conversion or decoding). After initial amplification by the LNA, the signal level may fluctuate due to varying reception conditions or circuit losses. VGA ensures amplitude remains within the optimal dynamic range, preserving signal quality. The Outport block outputs the processed signal—either in the form of demodulated data or intermediate results—for further analysis or integration into other subsystems.

Accurate demodulation in 5G receivers requires precise separation of the incoming RF signal into in-phase (I) and quadrature (Q) components. The proposed model enables programmable simulation of impairments affecting quadrature components, specifically evaluating the impact of I/Q gain mismatch, which causes signal constellation distortion, complicating symbol detection and degrading rejection of image frequencies. We also assessed the effects of I/Q phase mismatch, which results in rotation and displacement of the signal constellation, impairing accurate information recovery. For an RF demodulator, maintaining strict orthogonality between I and Q is critical. Any mismatch may lead to phase crossing and significantly reduce demodulation accuracy.

To mitigate the effects of both, gain and phase mismatches, we employed a combination of digital compensation techniques and calibration algorithms. Adaptive correction algorithms were implemented to estimate and compensate for these impairments based on real-time signal analysis. Specifically, we explored blind IQ correction techniques that rely on statistical properties of the signal to estimate deviations and apply corrective adjustments in the digital domain. Additionally, we investigated the use of pilot signals—known patterns introduced into the transmission—which help estimate I/Q channel parameters with higher precision and enable improved calibration. Finally, from a hardware implementation perspective, we consider the integration of compensation modules into field-programmable gate array (FPGA) or digital signal processor (DSP) platforms to be highly promising. Such integration enables real-time correction through a combination of hardware-assisted preprocessing and digital fine-tuning, making the receiver more robust under dynamic channel conditions.

The RF Transmitter Subsystem block models typical distortions [23], in particular, I/Q imbalance due to gain or phase mismatch between parallel signal conversion sections in the transmitter (will be discussed later in the article).

We represent the received signal with M-QAM modulation as follows:

$$r(t) = G_a \sum_k \left\{ a_0(k)p(t - kT_s)\cos(\omega_0 t + \Theta) - a_1(k)p(t - kT_s)\sin(\omega_0 t + \Theta) \right\} + w(t)$$

(11)

where $a_0(k)$ and $a_1(k)$ are the in-phase and quadrature components of the $k$-th symbol, $p(t)$ is the unit energy pulse shape, $T_s$ is the symbol duration time, $\omega_0$ is the center frequency of the received signal spectrum, $\Theta$ is the carrier phase offset, and $w(t)$ is additive white Gaussian noise. The constant $G_a$ represents all gains and losses in the antenna, propagation medium, amplifiers, mixers, filters, and other RF components. The received signal is sampled at a rate of $F_s = 1/T$ samples per second.

The signal at the receiver input is down-converted using quadrature sinusoids $\cos(\Omega_0 n + \Theta(n))$ and $-\sin(\Omega_0 n + \hat{\Theta}(n))$ obtained by the local oscillator.

Below (Fig 2) we present the demodulator circuit.

We would like to emphasize that RF to LO isolation is a parameter that determines how well the signal at the radio frequency is isolated from the signal at the local oscillator frequency in the mixer R_Isolation = -60 dB.

 

At the output of the mixers, we extract the in-phase and quadrature components, which, neglecting the double frequency components, can be expressed as:

$$I(nT) = G_a \sum_k \left\{ a_0(k)p(nT - kT_s)\cos(\Theta - \hat{\Theta}(n)) - a_1(k)p(nT - kT_s)\sin(\Theta - \hat{\Theta}(n)) \right\}$$
$$+ w_I(nT) \tag{12}$$

$$Q(nT) = G_a \sum_k \left\{ a_0(k)p(nT - kT_s)\sin(\Theta - \hat{\Theta}(n)) + a_1(k)p(nT - kT_s)\cos(\Theta - \hat{\Theta}(n)) \right\}$$
$$+ w_Q(nT) \tag{13}$$

The in-phase and quadrature components are filtered by a matched filter whose impulse response is $p(-nT)$. The mathematical models of the in-phase and quadrature signals at the output of the matched filter are:

$$x(nT) = \frac{G_a}{T} \sum_k \left\{ \left[ a_0(k)\cos(\Theta - \hat{\Theta}(n)) - a_1(k)\sin(\Theta - \hat{\Theta}(n)) \right] r_p(nT - kT_s) \right\} + v_I(nT) \tag{14}$$

$$y(nT) = \frac{G_a}{T} \sum_k \left\{ \left[ a_0(k)\sin(\Theta - \hat{\Theta}(n)) + a_1(k)\cos(\Theta - \hat{\Theta}(n)) \right] r_p(nT - kT_s) \right\} + v_Q(nT) \tag{15}$$

where $v_I(nT) = p(-nT) * w_I(nT)$, $v_Q(nT) = p(-nT) * w_Q(nT)$, $r_p(u)$ is the autocorrelation function of the pulse shaper, which can be represented as:

$$r_p(u) = \int_{-T_s}^{T_s} p(t)p(t - u)dt \tag{16}$$

Assuming perfect time synchronization, $x(nT)$ and $y(nT)$ are sampled at $n = kT_s/T = kN$ to obtain a signal space projection corresponding to the $k$-th symbol. When the pulse shape satisfies the Nyquist condition of no intersymbol interference, $r_p(0) = 1$ and $r_p(mT_s) = 0$ for $m \neq 0$, so that:

$$x(kT_s) = K \sum_k \left\{ \left[ a_0(k)\cos(\Theta - \hat{\Theta}(kN)) - a_1(k)\sin(\Theta - \hat{\Theta}(kN)) \right] \right\} + w_I(nT) \tag{17}$$

$$y(kT_s) = K \sum_k \left\{ \left[ a_0(k)\sin(\Theta - \hat{\Theta}(kN)) + a_1(k)\cos(\Theta - \hat{\Theta}(kN)) \right] \right\} + w_Q(nT) \tag{18}$$

where $K = G_a/T$.

## 2.3. Phase noise research

Now, let us consider the proposed methodology for evaluating phase noise. This methodology is highly relevant in the context of advancing modern 5G technologies, which demand precise methods for analyzing signal quality due to high modulation density, wide bandwidths, and stringent requirements for phase stability [24]. We emphasize that phase noise is one of the critical factors affecting the performance of OFDM systems, which form the foundation of 4G/5G standards.

 

Our approach accounts for the cumulative impact of phase noise on signal characteristics, including the EVM, and enables accurate modeling of its effect on system performance.

Unlike existing studies, which predominantly analyze the impact of phase noise in isolation or with simplified models, the proposed methodology integrates multiple factors, including overall phase shift, ICI, and spectral distortions caused by signal power broadening. This ensures a comprehensive approach to evaluating phase noise and its implications for next-generation communication systems.

The methodology for assessing phase noise is presented below, based on a detailed analysis of the physical processes causing this phenomenon and their impact on signal quality in OFDM systems.

Phase noise occurs when the phase of a high frequency oscillator has rapid, short-term random phase variations. Fig 6 shows an ideal local oscillator with a Dirac spectrum compared to a real PN oscillator.

We emphasize that PN is determined by the frequency region in the 1 Hz band with an offset f from the carrier frequency. The PN power in this band is normalized relative to the carrier power, dBc/Hz. For example, in GSM generators, PN should fall below 115 dBc/Hz when restoring 600 kHz [3].

PN causes abrupt changes in the frequency spectrum and time characteristics of the output signal generator. In particular, PN broadens the power spectral density (PSD) on both sides of the signal (Fig 6b), which causes adjacent channel interference (ACI).

Phase noise is a random process and causes three main problems in OFDM systems: overall phase shift, power degradation, and ICI [25,26]. Considering an OFDM signal that is only affected by phase noise ($\varphi(n)$) at the receiver, the resulting time domain signal can be expressed as:

$$y_m(n) = x_m(n)e^{j\varphi_m(n)} \tag{19}$$

Assuming that the phase shift is small (i.e., $e^{j\varphi_m(n)} \approx 1 + j\varphi(n)$) the reconstructed symbols are expressed in the following form [27]:

$$Y_m(k) \approx X_m(k)\left\{1 + j\frac{1}{N}\sum_{n=0}^{N-1}\varphi(n)\right\} + \frac{j}{N}\sum_{r=0}^{N-1}X_m(r)\sum_{n=0}^{N-1}\varphi_m(n) \cdot e^{j(2\pi/N)(r-k)n} \tag{20}$$

Note that the general phase term (18) introduces rotation into the constellation. This rotation is the same for all subcarriers and is representative of the average PN. When the phase noise is small, the overall phase format is an effect of the dominant phase noise, and it accumulates over time (i.e., the variance increases over time). The problems associated with this

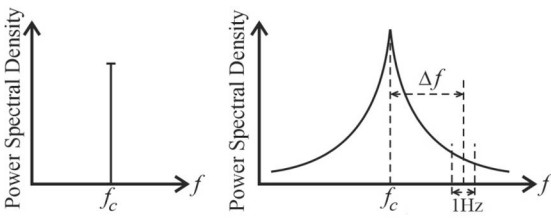

**Fig 6. Power spectral density: a) of an ideal heterodyne; b) of a real heterodyne.**

term can be easily avoided by carefully implemented pilot tracking [28]. The last term (18) represents the leakage from adjacent subcarrier frequencies (i.e., ICI) and is shown in Fig 7.

This situation cannot be corrected because the phase offset ($\varphi_m(n)$) and the input data sequence ($X_m(n)$) are random. Thus, it leads to a deterioration of the SNR in the overall system. The only way to reduce the interference due to this case is to improve the generator performance with an associated increase in accuracy. The ICI power via phase noise can be roughly modeled as follows [26,29,30]:

$$P_{ICI} \cong \frac{\pi \beta T_s E_s E_H}{3}$$

(21)

where $\beta$ represents the two-sided 3 dB bandwidth (i.e., the frequency ranges between the 3 dB points of its Lorentzian power spectral density [31]), $T_s$ denotes the OFDM symbol duration [32], $E_s$ and $E_H$ represent the symbol energy and channel power, respectively, which are often normalized to unity. Consequently, the signal-to-phase-noise-plus-interference ratio (SPNIR) can be expressed as follows [26]:

$$SPNIR = \frac{1 - \frac{\pi \beta T_s}{3}}{\frac{\pi \beta T_s}{3}}$$

(22)

The numerator in the above equation accounts for the degradation of signal power due to PN. It is important to note that Equation (22) assumes the absence of any other noise or interference in the system.

Based on the presented methodologies, we will determine the prospects for their theoretical and practical implementation. The proposed phase noise assessment techniques demonstrate exceptional relevance and practical significance in the context of contemporary challenges faced by telecommunications systems, electronics, and radio engineering. Specifically, with the development of 5G technologies, which are characterized by high modulation density, wide bandwidths, and increased demands on phase stability, the accuracy of phase noise estimation becomes critical for ensuring the reliability and performance of communication systems.

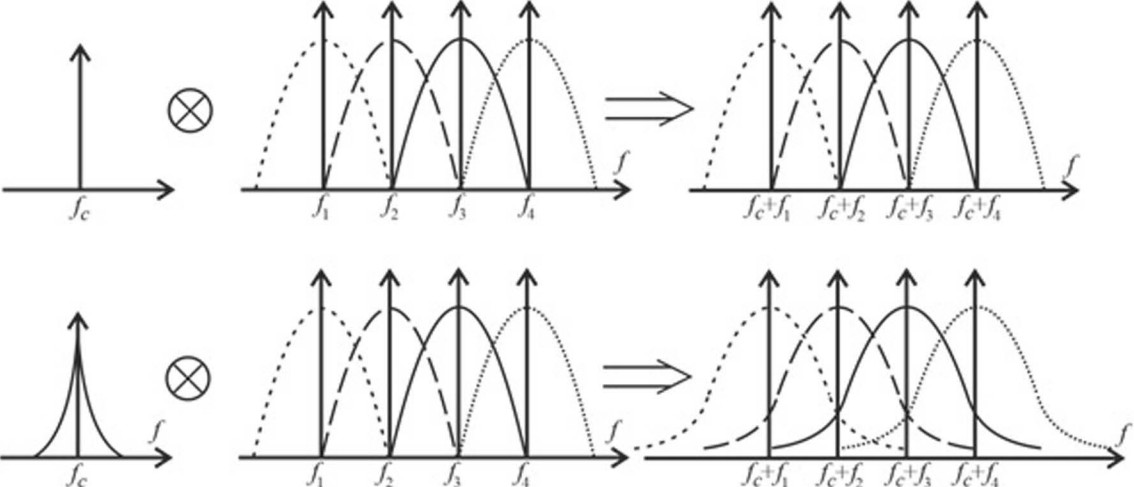

**Fig 7. Effect of PN on OFDM subcarriers: a) ideal heterodyne; b) practical heterodyne.**

Therefore, the approach developed in this article offers a comprehensive analysis of the impact of phase noise, integrating aspects such as overall phase shift, ICI, and spectral distortion. This level of detail enables effective assessment of real-world scenarios encountered in next-generation radio technologies and suggests pathways for their improvement. Thus, the methodologies have both theoretical and practical value, as they contribute to the development of new approaches for improving the performance of OFDM systems that underpin modern communication standards.

The significance of the presented methodologies also lies in their ability to account for complex physical processes that affect signal quality. This creates a foundation for improving RF interfaces in emerging technologies, allowing developers to more accurately predict system performance and make informed decisions when designing RF front-end circuits.

The next section will present experimental results that detail the practical performance metrics within the scope of our research. We are confident that the results provided below will not only validate the developed methodologies but also offer practical recommendations for developers on optimizing the RF front end in line with the modern requirements of 4G/5G systems, which is crucial for the practical implementation of the results.

## 3. Results

In this section, we present the results of research aimed at analyzing and evaluating the characteristics of RF transmitters and receivers for LTE and 5G systems using models developed in the Simulink environment. A key feature of this study is a detailed examination of the impact of RF impairments, such as phase and amplitude imbalance, phase noise, and amplifier nonlinearities, on the system's performance metrics. The relevance of the obtained results lies in their potential to optimize transmitter and receiver performance, enhance their characteristics, and ensure compliance with modern communication standards. The presented research can be applied in the design and testing of equipment for next-generation mobile communication systems. To provide a comprehensive overview of the characteristics and specific aspects of the conducted research, this section includes the following research result: Analysis and investigation of the spectral characteristics of signals at the transmitter output; Evaluation of the impact of I/Q imbalance and amplifier nonlinearities on signal quality, represented through EVM; Performance assessment of an LTE RF receiver using the RMC; Analysis of the dependencies between BER, SNR, and various system parameters; Sensitivity analysis of the LTE receiver de-pending on bandwidth, carrier frequency, and modulation type. We emphasize that each of these experimental and research aspects will be illustrated with corresponding graphs, structural diagrams, and detailed simulation results. A qualitative analysis and proposed discussion of the findings from this section, as well as the overall study, will be provided in the subsequent Discussion section.

The Discussion section will provide a comprehensive analysis of the experimental results, emphasizing the impact of critical hardware parameters on the signal quality of radio receivers in LTE and 5G systems. It will highlight the sensitivity of EVM to IQ imbalance, amplifier nonlinearity, and phase noise, reaffirming their importance in high-speed telecommunication systems.

The QM78207 chipset, manufactured by Qorvo [33], was selected as the core component for our research due to its characteristics, which enable a deeper analysis of radio frequency distortions in systems utilizing 64-QAM modulation. Key parameters of this chipset include support for a wide frequency range (from 50 MHz to 6 GHz), low power consumption, and high accuracy in real-time distortion analysis. These features allow for a detailed examination of the impact of distortions on signal transmission performance in complex radio frequency environments.

### 3.1. Evaluation of spectral characteristics and I/Q imbalance

Let's consider the Simulink model of a digital communication system for LTE and 5G, as shown in Fig 8.

We will investigate the impact of radio frequency distortions, including in-phase and quadrature imbalance, PN, and PA nonlinearities, on the performance of the LTE and 5G RF transmitter.

The signal spectrum at the transmitter output, as displayed by the Spectrum Analyzer, is shown in Fig 9.

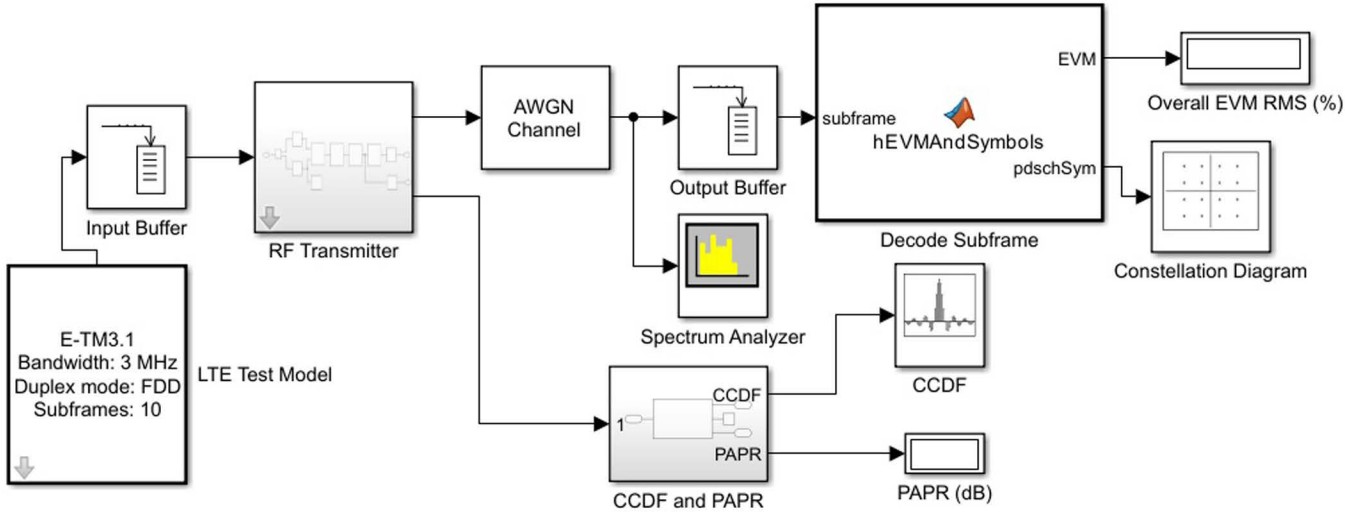

**Fig 8. The Simulink model of the LTE and 5G digital communication system.**

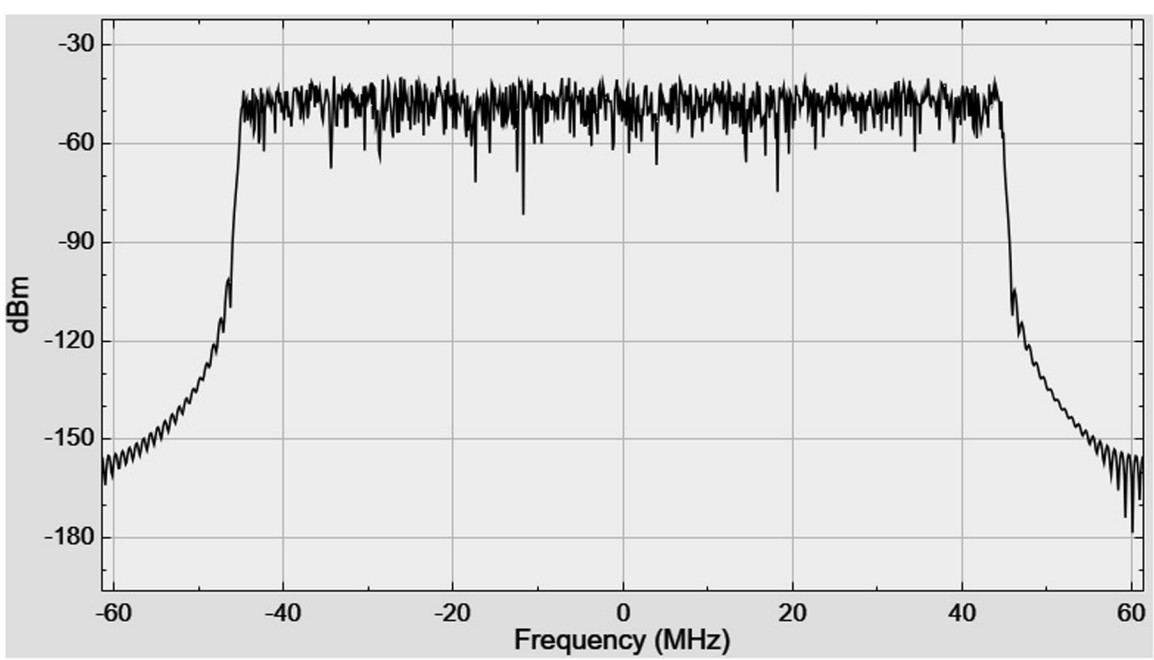

**Fig 9. The signal spectrum at the transmitter output.**

The 64-QAM constellation [14,34] at the transmitter output is shown in Fig 10a. Fig 10b presents the 64-QAM constellation in the presence of imbalance: I/Q gain mismatch = 3 dB; I/Q phase mismatch = 10°.

To analyze intermodulation distortions, a Fourier analysis is required to identify different components at the output, which necessitates the output signal being periodic. For this reason, both input tones must share a common period. Typically, the tones are very close in frequency, implying that the common period will be relatively long compared to the period of any individual tone. This means that modeling transient processes will be computationally expensive.

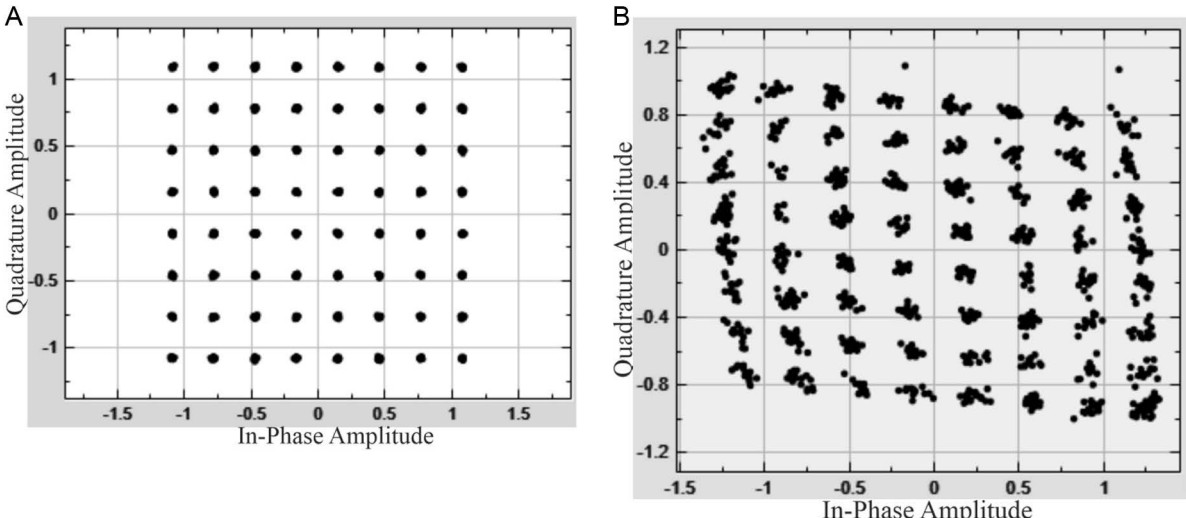

**Fig 10. 64-QAM constellation: a) at the transmitter output; b) in the presence of imbalance: I/Q gain mismatch = 3 dB; I/Q phase mismatch = 10°.**

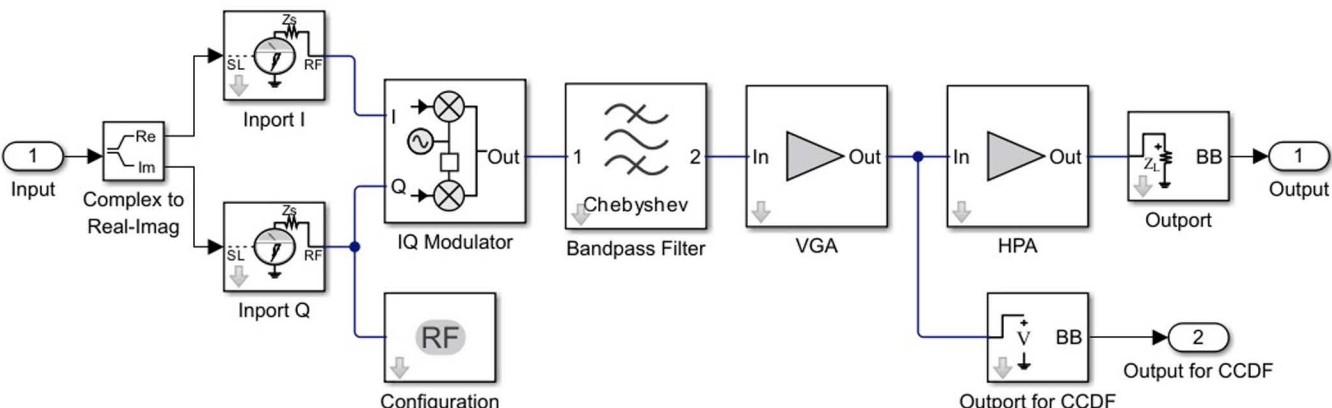

**Fig 11. Simulink model of the RF transmitter block.**

In Fig 11, we present the Simulink model of the RF Transmitter block.

In Fig 12, we present the synthesized equivalent structural diagram of the Simulink model of the RF Transmitter block.

The RF Transmitter Subsystem block consists of the following components: IQ modulator, bandpass filter, and power amplifier [24,35]. The transmitter includes an amplifier with a VGA to control the input back-off (IBO) level of the high-power amplifier (HPA).

Next, we conduct a study of the radio frequency receiver for the downlink according to the Simulink model in Fig 13.

The proposed model includes the following components and simulation configuration parameters: Bandpass filter. We will define the filter coefficients for the Simulink model. The filter has an order of 32, with a passband frequency of 2.25 MHz and a stopband frequency of 2.7 MHz; SNR and signal energy; $Noc$ is the spectral power density of the white noise source; SNRdB = 57 ($Es/Noc$ in dB); $Noc$dBm = -98 ($Noc$ in dBm/15kHz); $Noc$dBW = $Noc$dBm - 30 (-128 $Noc$ in dBW/15kHz); SNR = 10^(SNRdB/10) (5e5 linear $Es/Noc$); $Es$ = SNR*(10^($Noc$dBW/10)) (7,9e-8 linear $Es$ per RE); FFTLength = 512; SymbolPower = $Es$/double(FFTLength) (1,55e-10); Number of frames: $N$ = 3.

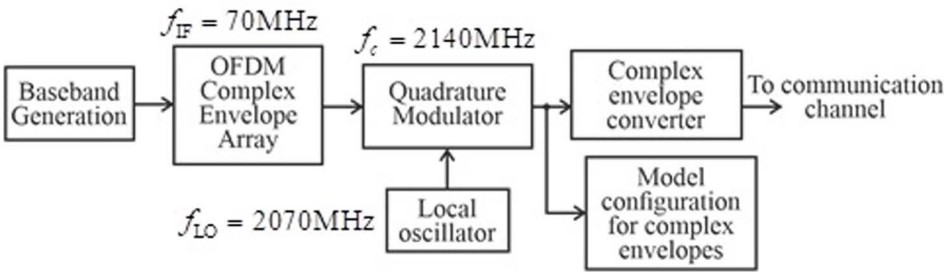

**Fig 12. The equivalent structural diagram of the Simulink model of the RF Transmitter block.**

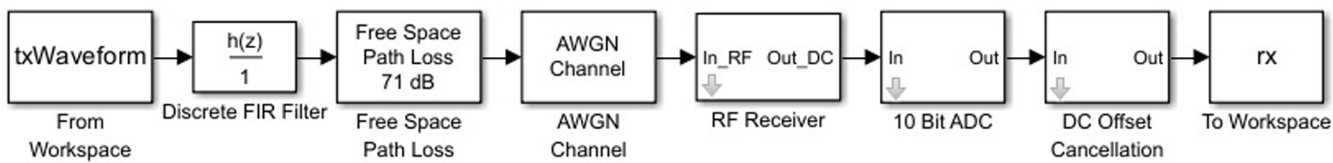

**Fig 13. Simulink model of the radio frequency receiver for the LTE downlink.**

In Fig 14, we present the graphical representation of the filter coefficients for the bandpass filter [36].

First, the test signal is generated. The Reference Measurement Channel (RMC) in LTE is used for signal synchronization and initial distortion correction. The primary goal of using the RMC for receivers is to provide a stable, controlled, and repeatable testing environment.

The downlink signal consists of the following components. The RMC has the following parameters: 25 resource elements are used; modulation: 64-QAM; code rate: 3/4 [5].

Now, let's define the frame structure type for the duplex mode – FDD (Frequency Division Duplex). In LTE, information transmission is organized into radio frames with a duration of 10 ms, which consist of 10 consecutive subframes, each made up of several consecutive OFDM symbols [37], as shown in Fig 15 [24].

Here, we used FDD – a method employed for communication using two different radio frequencies for transmission and reception, separated by a specific guard interval [38].

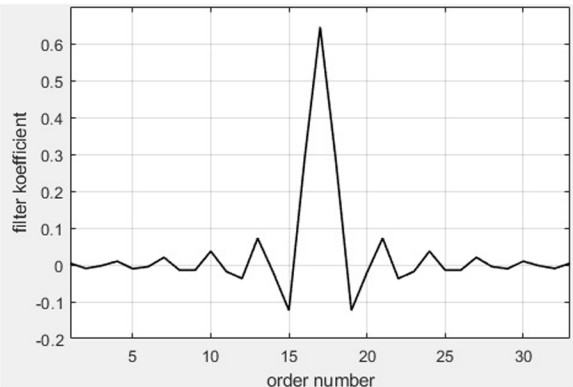

**Fig 14. Graphical representation of the filter coefficients for the bandpass filter.**

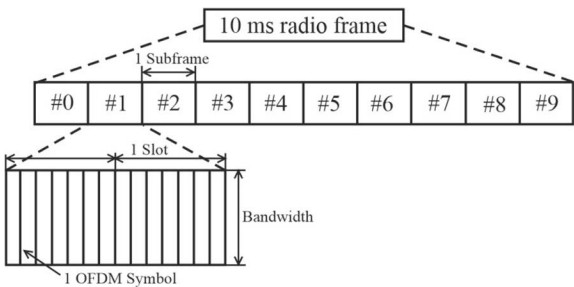

**Fig 15. LTE radio frame structure.**

Next, we present the experimentally obtained Bit Error Rate (BER) versus Signal-to-Noise Ratio (SNR) dependencies for the LTE Physical Downlink Shared Channel with different modulation schemes [23]: 1 is the QPSK, 2 is the 16QAM, 3 is the 64QAM, 4 is the 256QAM (see Fig 16).

Once again, we emphasize that we used the RMC for LTE to test the receiver in order to evaluate its performance.

We employed the following key metrics for receiver testing: receiver sensitivity, receiver interference immunity, performance evaluation under different duplex modes (FDD/TDD), testing signal recovery capabilities under multipath propagation conditions, receiver testing with varying bandwidths, and performance assessment of PDSCH (Physical Downlink Shared Channel) decoding.

Next, we present the obtained dependence of the LTE receiver sensitivity with a 10 MHz bandwidth and various modulation schemes on the SNR, as shown in Fig 17.

The obtained dependence of LTE receiver sensitivity with a 10 MHz bandwidth on the carrier frequency is shown in Fig 18. Additionally, the dependence of LTE receiver sensitivity with a 10 MHz bandwidth on the SNR is shown in Fig 19.

From the obtained dependencies, the following conclusions can be drawn. Specifically, as the bandwidth of the communication channel increases, the noise power rises, leading to a degradation in receiver sensitivity. Higher-order modulation schemes require increased SNR for accurate signal reception due to the higher symbol density. The extended cyclic prefix (OFDM) [39] provides better protection against multipath propagation but slightly reduces the spectral efficiency of the communication system.

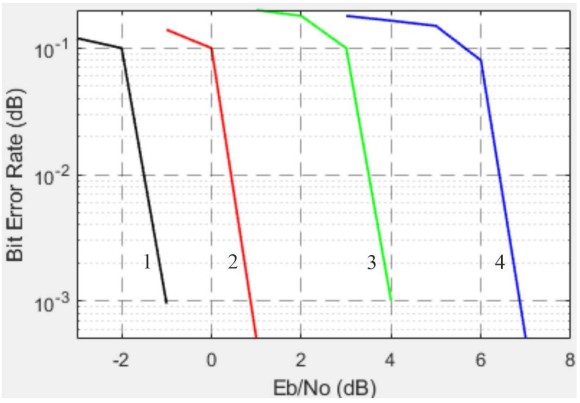

**Fig 16. BER vs. SNR for the Physical Downlink Shared Channel (LTE) with different modulation schemes: 1 is the QPSK, 2 is the 16QAM, 3 is the 64QAM, 4 is the 256QAM [18].**

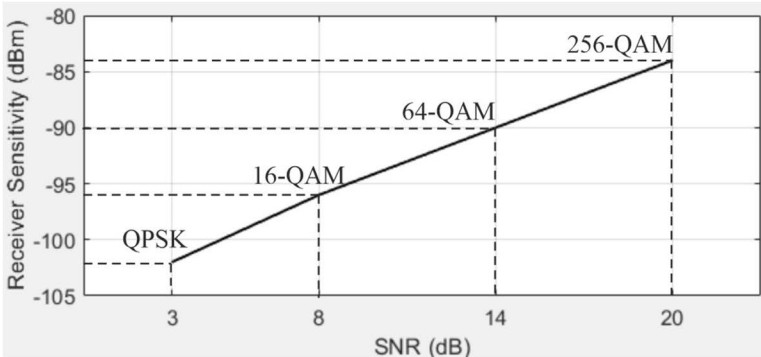

**Fig 17. Dependence of LTE receiver sensitivity with a 10 MHz bandwidth and various modulation schemes on the SNR.**

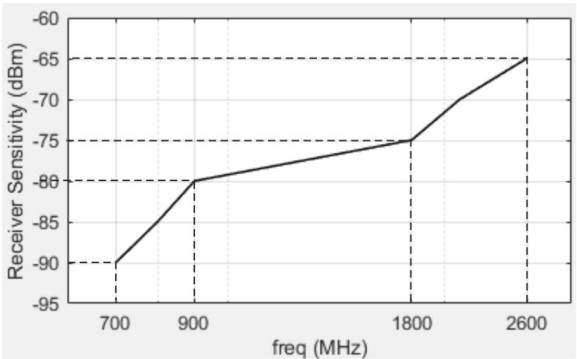

**Fig 18. Dependence of LTE receiver sensitivity with a 10 MHz bandwidth on the carrier frequency.**

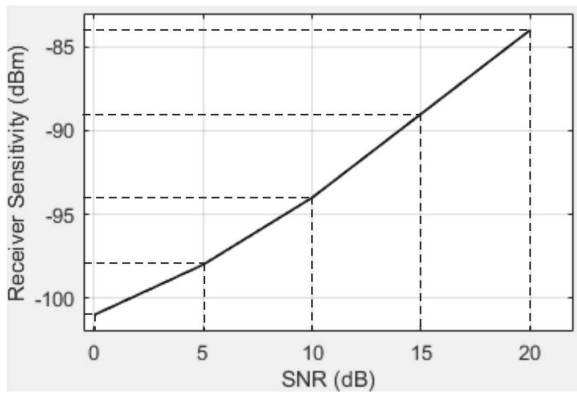

**Fig 19. Dependence of LTE receiver sensitivity with a 10 MHz bandwidth on the SNR.**

Next, we present a graphical representation of the occurrence of amplitude and phase imbalance in a quadrature modulator, shown in Fig 20a. The dependence of the error vector value obtained in the study on SNR for different values of I/Q imbalance is shown in Fig 20b.

The dependence of the EVM on SNR for different I/Q imbalance phases is shown in Fig 21.

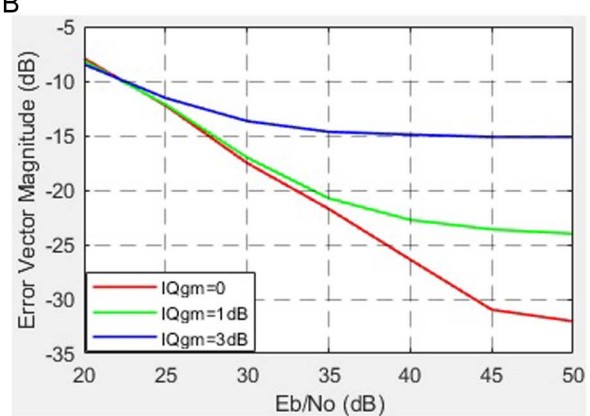

**Fig 20. Towards the assessment of phase imbalance in a quadrature modulator: a) graphical representation of the occurrence of amplitude and phase imbalance in a quadrature modulator; b) EVM versus SNR for different I/Q imbalance values.**

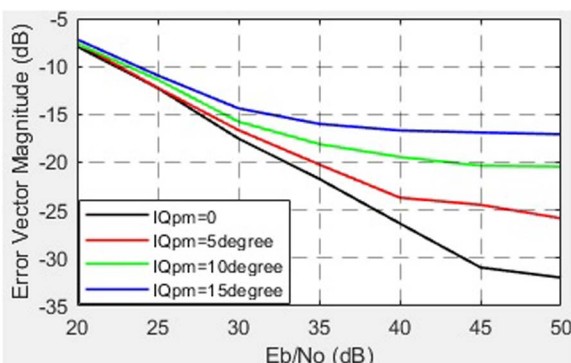

**Fig 21. The dependence of the EVM on SNR for different I/Q imbalance phases.**

### 3.2. Reception and measurement of baseband signals

As a result of our research, we present an assessment of the reception and measure-ment of baseband signals.

The Decode Subframe block performs OFDM demodulation [39,40] of the received subframes, channel estimation and equalization to reconstruct and plot the PDSCH symbols in the constellation diagram. EVM [34] is measured at two points in time (low and high), where the low and high locations correspond to the fast Fourier transform (FFT) window [25] align-ment at the beginning and end of the cyclic prefix.

The dependence of the EVM on the SNR for different values of the third-order nonlinearity ($IP_3$) of VGA is shown in Fig 22.

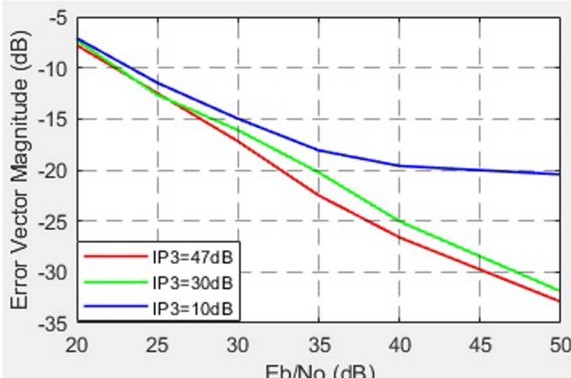

**Fig 22. Dependence of the EVM on the SNR for different values of the third-order nonlinearity ($IP_3$) of VGA.**

The dependence of the EVM on the SNR for different values of the third-order nonlinearity ($IP_3$) of the HPA is shown in Fig 23.

The amplifier maintains a constant gain for low-level input signals. However, at higher input signal levels, the amplifier saturates and its gain decreases. The 1 dB gain compression point ($P_{1dB}$) indicates the power level that results in a gain drop of 1 dB from the small-signal (linear amplifier) value.

$P_{1dB}$ is the power at the 1 dB compression point, usually used as a reference when choosing the IBO HPA level. Let us analyze the case where IBO = 12 dB, corresponding to the HPA operating in the linear region, and compare it with the case where IBO = 6 dB, corresponding to the HPA starting to operate in the nonlinear region. The VGA gain controls the IBO level. To maintain the linear behavior of the VGA, it is necessary to choose a gain value below 20 dB.

The linear PA has an IBO of 12 dB. To operate at an IBO of 12 dB, the gain VGA = 0 is shown in Fig 24.

The nonlinear HPA has an IBO of 6 dB. To operate at an IBO of 6 dB, the gain VGA = 6 dB. The dependence of the channel power for different gains of the variable gain amplifier on the SNR is shown in Fig 25.

According to the standards, the maximum EVM for a 64 QAM constellation is 8%. Since the overall EVM is about 1.7%, this architecture meets the standards.

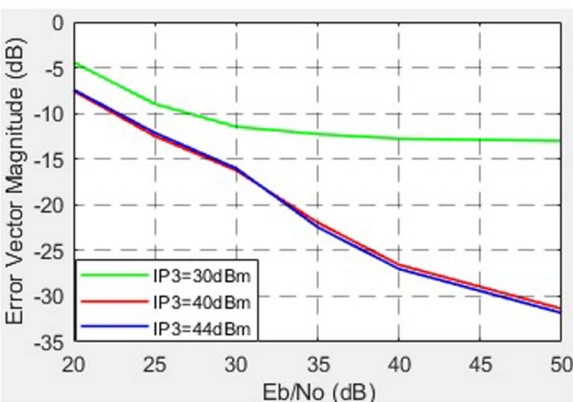

**Fig 23. Dependence of the EVM on the SNR for different values of the third-order nonlinearity ($IP_3$) of HPA.**

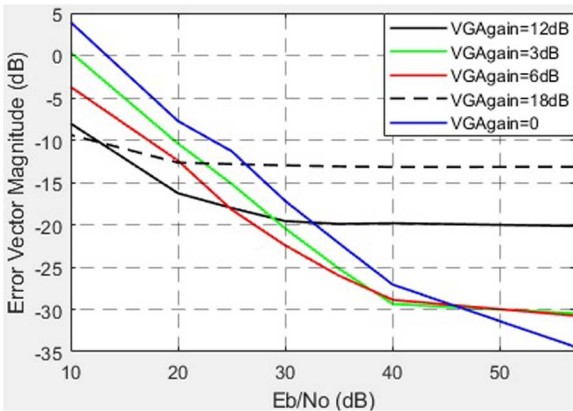

**Fig 24. EVM vs. SNR for different VGA gains.**

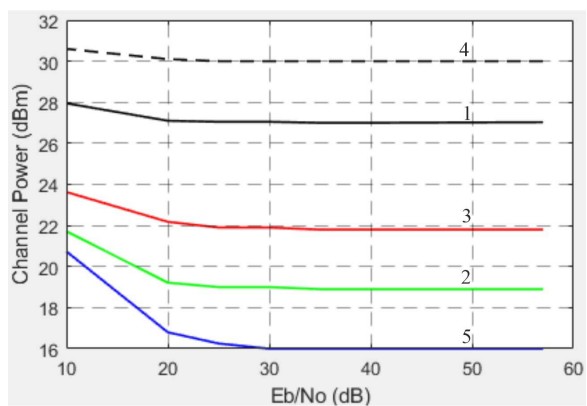

**Fig 25. Channel power for different VGA gains versus SNR: 1 is the VGA gains = 12 dB, 2 is the VGA gains = 3 dB, 3 is the VGA gains = 6 dB, 4 is the VGA gains = 18 dB, 5 is the VGA gains = 0.**

### 3.3. Phase noise research

The obtained dependence of the PN level on the PN offset is presented in Fig 26.

The dependence of EVM on SNR at different phase noise levels is shown in Fig 27.

Due to accuracy limitations, local oscillators in user equipment (UE) typically have a frequency stability tolerance of ±10 ppm. An LTE (5G) oscillator operating at 2.5 GHz with a stability of ±10 ppm exhibits a frequency offset of ±25 kHz.

## 4. Discussion

The results presented in this study highlight the influence of key hardware parameters of radio receivers on signal modulation quality, measured through the EVM metric. The identified patterns reaffirm the well-known sensitivity of communication systems to IQ imbalance, amplifier nonlinearity, and PN—factors that are critical for modern tele-communication systems, especially in the context of high-speed standards such as LTE and 5G.

The investigation of IQ amplitude imbalance demonstrates that even small deviations (0–3 dB) significantly degrade EVM (from -32 dB to -15 dB at SNR = 50 dB). These findings align with prior studies emphasizing the importance of precise IQ correction to ensure reliable signal transmission in wideband communication systems. Similarly, IQ phase

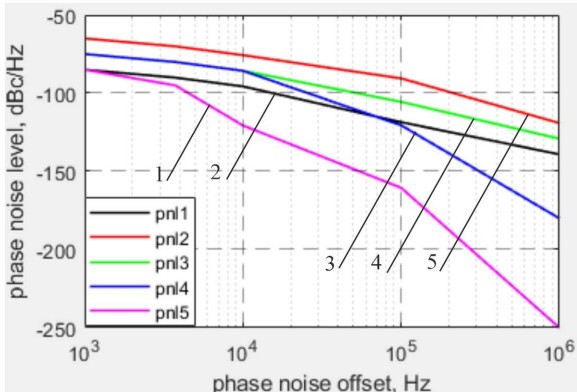

**Fig 26. Graphs showing the dependence of PN level on PN offset for local oscillators with different frequency stability.**

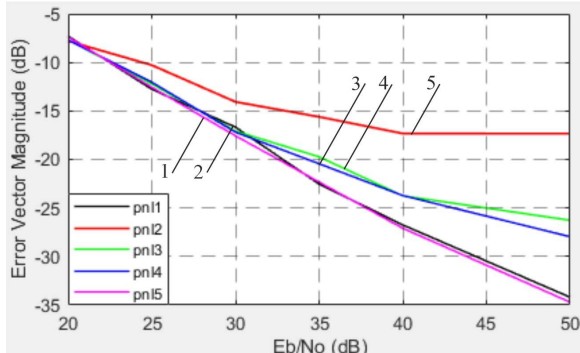

**Fig 27. Graphs of the dependencies of EVM on SNR for local oscillators with different frequency stability and phase noise levels.**

imbalance within the range of 0–15° shows a comparable trend, increasing EVM from -32 dB to -17 dB. This behavior can be attributed to increased ICI caused by phase distortion, which is particularly critical in systems with high-order modulation schemes, such as 64-QAM or 256-QAM [41].

Another crucial factor is amplifier nonlinearity. For low-power amplifiers (VGA), the results reveal that reducing the third-order intermodulation point ($IP_3$) from 47 dB to 10 dB significantly increases EVM (from -33 dB to -20 dB). For HPA, the findings confirm that allowable levels of nonlinearity are substantially lower: reducing $IP3$ from 44 dB to 30 dB results in EVM degradation from -32 dB to -13.5 dB (at SNR = 50 dB). These observations are consistent with modern trends requiring linear operation to minimize distortions in high-data-rate systems. The dependence of EVM on the VGA operational mode is also noteworthy. In the linear mode (gain up to 6 dB), EVM remains within acceptable limits (-35 to -30 dB), while the channel power increases. However, in the nonlinear mode (gain of 12–18 dB), EVM exceeds acceptable thresholds, underscoring the need to limit VGA operation to linear regions in high-performance systems.

The analysis of phase noise effects demonstrated that improving local oscillator stability significantly enhances EVM. At the permissible phase noise level of -120 dBc/Hz (1e5 Hz offset from the carrier), EVM reaches a critical level of -17.5 dB. Further improvements in stability to -160 dBc/Hz lead to an EVM improvement of up to -35 dB. This underscores the importance of local oscillator stability in systems with stringent signal quality requirements, such as 5G NR [42,43] and massive MIMO [44].

Based on the obtained results, we present an analysis of the connections between the results and the real-world challenges of 4G/5G networks. Specifically, the results confirm that IQ imbalance (both amplitude and phase) is a critical factor in ensuring signal quality, particularly when using high-order modulation schemes such as 64-QAM or 256-QAM. In real 5G systems, which support massive connectivity and high data rates, even a small IQ imbalance can lead to significant degradation of the EVM, thereby increasing the likelihood of data transmission errors. This issue is especially relevant for subsystems with wideband channels (>100 MHz), where the impact of imbalance is considerably amplified. Additionally, real-time IQ imbalance correction presents a significant challenge for modern chipsets, requiring substantial computational power without com-promising throughput. We would also like to highlight that the amplifier nonlinearity issue, as confirmed by our research, is critical for real RF interfaces in 4G/5G systems. In 5G networks, which employ massive MIMO arrays and beamforming technologies, amplifier nonlinearity can result in the generation of spurious frequency components, reducing spectral efficiency. This is especially important in high-density user environments, where maintaining a stable signal for each user is essential. Therefore, the data presented in this study underscore the impor-tance of operating amplifiers in linear modes to maintain the required EVM levels, which should be considered when designing RF chains. Furthermore, the impact of phase noise on EVM demonstrates that LO stability is one of the most important parameters for ensuring signal quality in 5G. For example, in massive MIMO and mmWave systems, where pre-cise synchronization is crucial, phase noise can significantly degrade the performance of beamforming algorithms. Addi-tionally, the influence of phase noise increases substantially with the use of high-order modulation, which is characteristic of modern networks with high data rate demands.

From the perspective of linking the results of this study to the real needs of 4G/5G networks, we emphasize that the analysis of these relationships indicates that the findings can be directly applied to optimize RF design in modern 4G/5G networks. For instance, modeling the impact of key parameters can be used to prevalidate the performance of RF chipsets such as the QM78207, which will contribute to the development of more reliable devices that meet current signal quality requirements. Finally, we would like to address the specifics of 5G system design and optimization. Our research could serve as a foundation for developing real-time distortion compensation algorithms, such as adaptive IQ imbalance or phase noise correction. This is particularly relevant for systems with high mobility (e.g., transport or drone applications), where signal parameters can change dynamically.

The findings of this study are crucial for optimizing the parameters of modern radio receivers operating in challenging environments characterized by multipath propagation and high levels of interference. Future research could focus on devel-oping IQ imbalance correction algorithms, enhancing local oscillator stability, and implementing adaptive control of amplifier operating modes based on network conditions. Moreover, future research will encompass and focus on the use of artifi-cial intelligence (AI) and machine learning (ML) techniques for the real-time prediction and correction of aberrations in RF systems. Machine learning models can be trained on datasets that include measurements of IQ imbalance, phase noise, and nonlinearity to dynamically adjust the settings of RF systems. This will significantly improve system performance by automatically adjusting its parameters based on real-time measurements and minimizing the impact of aberrations that can degrade signal quality. One of the key objectives is to integrate such technologies in real-time, enabling systems to quickly adapt to changing conditions and enhance their stability in high-speed data transmission environments. This approach will allow the adaptive adjustment of system parameters to ensure stable operation under various influences, such as chan-nel changes, load variations, or environmental factors. The implementation of AI and ML in RF systems will open up new opportunities for enhancing the performance and reliability of modern communication technologies. Our research will focus on integrating massive MIMO and 256-QAM technologies, which are crucial for next-generation systems. Future work may explore correction algorithms and adaptive tuning methods for systems with large antenna arrays and high-order modula-tion, as well as advancements in oscillator stability and power amplifier linearity to improve signal quality.

In this context, we would like to highlight certain limitations of the presented study and outline potential areas for our future research. Despite the valuable insights obtained, this study has several limitations that should be addressed in future

research. First, the analysis was conducted under controlled laboratory conditions, which may not fully capture the complex effects of real-world interference, varying environmental factors, and hardware imperfections encountered in practical deployments. Future studies should incorporate field measurements and dynamic channel conditions to validate the findings in real-world 5G network scenarios, thereby improving the applicability of these results in operational environments.

Second, while the study focuses on key hardware impairments such as IQ imbalance, amplifier nonlinearity, and phase noise, other factors, including mutual coupling in massive MIMO arrays and power amplifier memory effects, may further impact signal quality. Investigating these additional impairments would provide a more comprehensive understanding of RF system behavior, and the interplay between various hardware distortions in real-world systems remains an area requiring further exploration.

Moreover, this study primarily evaluates EVM as a performance metric. While EVM is a critical indicator of signal quality, it does not account for higher-layer performance metrics, such as BER and Frame Error Rate (FER), which are essential for assessing end-to-end system performance. Future research should investigate the correlation between EVM and these metrics to better understand the practical implications of RF impairments on overall network reliability and user experience.

In addition, this study highlights the potential of AI-driven adaptive algorithms for real-time compensation of RF impairments, but further investigation is needed to develop and validate machine learning-based models capable of dynamically adjusting RF parameters in response to changing network conditions. The practical integration of such solutions into 5G and beyond-5G networks remains an open challenge, with many technical and operational hurdles to overcome.

Lastly, as the field of communication technologies advances toward 6G, emerging technologies such as ultra-massive MIMO, terahertz communications, and reconfigurable intelligent surfaces (RIS) will introduce new challenges related to RF impairments. Future research should explore how the findings of this study can be extended to next-generation systems to ensure the continued enhancement of signal integrity and spectral efficiency in the face of these novel technologies.

## 5. Conclusions

Based on the conducted research, the following conclusions can be drawn. When the IQ amplitude imbalance varies from 0 to 3 dB, the EVM increases from -32 dB to -15 dB at an SNR of 50 dB. A change in the IQ phase imbalance from 0° to 15° leads to an increase in EVM from -32 dB to -17 dB at an SNR of 50 dB. Analysis of EVM dependency on third-order nonlinearity ($IP3$) for VGA shows that a reduction in $IP3$ from 47 dB to 10 dB results in EVM degradation from -33 dB to -20 dB. For HPA, the permissible level of nonlinearity is lower: a decrease in IP3 from 44 dB to 30 dB causes EVM to worsen from -32 dB to -13.5 dB at an SNR of 50 dB. Regarding EVM and the operational mode of VGA, the linear mode offers the best performance. When the VGA gain increases from 0 to 6 dB, the EVM changes from -35 dB to -30 dB, remaining within acceptable limits. Simultaneously, the channel power increases from 16 dB to 22 dB at an SNR of 57 dB. In the nonlinear mode of VGA operation (gain levels of 12 dB and 18 dB), the EVM increases to -20 dB and -12 dB, respectively, exceeding acceptable thresholds. Concurrently, the channel power rises from 27 dB to 30 dB at an SNR of 57 dB. The analysis of EVM as a function of phase noise levels reveals that, at the standard-permitted phase noise level of -120 dBc/Hz (offset of 1e5 Hz from the carrier), the worst EVM value of -17.5 dB is observed with a phase noise level of -90.6 dBc/Hz. Further improvement in local oscillator stability to -160 dBc/Hz (at the same offset) reduces the EVM to -35 dB.

The main set of practical recommendations and potential applications of the research findings presented in this paper is as follows. Regarding IQ imbalance compensation, we propose utilizing automatic adjustment schemes to address amplitude (up to 3 dB) and phase (up to 15°) imbalance. This can be achieved through digital compensators integrated into the RF chain. Additionally, implementing adaptive calibration algorithms for IQ imbalance during system startup is highly relevant. For improving LO stability, we emphasize the importance of maintaining phase noise levels of no worse than -120 dBc/Hz (at a 100 kHz offset) for medium-precision systems and up to -160 dBc/Hz for high-performance systems like 5G NR. Adopting technologies such as direct digital synthesis (DDS) or highly stable quartz oscillators can effectively meet these requirements. Concerning the minimization of PA nonlinearity, we recommend employing PAs with high $IP3$

values exceeding 40 dB for VGA and 30 dB for HPA. Furthermore, linearization techniques such as pre-compensation (e.g., Doherty amplifiers), digital predistortion (DPD), and limiting amplifier operation to the linear gain region (up to 6 dB for VGA) are both necessary and appropriate. To mitigate channel noise effects, we recommend adaptive filtering algorithms that account for signal spectral characteristics, along with optimizing SNR during the design phase using efficient error correction codes (e.g., LDPC or Polar Codes, as demonstrated in our prior works [5,6,42]). Transitioning to more robust modulation schemes is another avenue for improvement. Moreover, for critical channels, QPSK can be employed, albeit at the cost of reduced data rates. Implementing adaptive modulation schemes that dynamically adjust to channel conditions can further reduce distortion impact, as highlighted in our findings in [45]. Finally, we stress the importance of adopting integrated solutions in practical implementations. Specifically, as outlined in the methodology of this paper, using integrated chipsets (such as the QM78207, chosen as the baseline for this study) that combine high-precision RF components with digital blocks for distortion monitoring and correction is highly recommended. Implementing real-time EVM monitoring at the hardware level will enable immediate response to critical changes in the RF chain. The results and recommendations provided in this paper will facilitate accurate and efficient RF interface tuning to minimize signal distortions. Moreover, we highlight the necessity of thorough RF interface testing in accordance with 3GPP standards for 4G and 5G (LTE, NR), including assessments of EVM, SNR, and other critical parameters. This comprehensive approach will aid in developing universal guidelines for telecommunications system designers. The findings of this study will ensure stable performance of 4G/5G systems under increasing demands for communication quality and network throughput.

## Acknowledgments

The authors are grateful to all persons and organizations that contributed to the publication of the article.

## Author contributions

**Conceptualization:** Ilya Pyatin.

**Data curation:** Oksana Kovtun.

**Formal analysis:** Ilya Pyatin, Juliy Boiko.

**Funding acquisition:** Viacheslav Kovtun, Oksana Kovtun.

**Methodology:** Ilya Pyatin, Viacheslav Kovtun.

**Project administration:** Viacheslav Kovtun.

**Resources:** Oksana Kovtun.

**Software:** Juliy Boiko.

**Supervision:** Viacheslav Kovtun.

**Validation:** Viacheslav Kovtun, Oksana Kovtun.

**Visualization:** Ilya Pyatin, Juliy Boiko.

**Writing – original draft:** Ilya Pyatin, Juliy Boiko.

**Writing – review & editing:** Ilya Pyatin, Juliy Boiko.

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
