## [Decision Letter · Decision Letter 0]

30 Mar 2025

PONE-D-25-06441Radio Frequency Interface Quality Assessment in 4G/5G: Effects of IQ Imbalance, Phase Noise, and Nonlinearities on Error Vector MagnitudePLOS ONE

Dear Dr. Kovtun,

Thank you for submitting your manuscript to PLOS ONE. After careful consideration, we feel that it has merit but does not fully meet PLOS ONE’s publication criteria as it currently stands. Therefore, we invite you to submit a revised version of the manuscript that addresses the points raised during the review process.

We look forward to receiving your revised manuscript.

Kind regards,

Sushank Chaudhary, Ph.D

Academic Editor

PLOS ONE

Journal Requirements:

Reviewers' comments:

Reviewer's Responses to Questions

**Comments to the Author**

1. Is the manuscript technically sound, and do the data support the conclusions?

Reviewer #1: Yes

Reviewer #2: Yes

2. Has the statistical analysis been performed appropriately and rigorously? 

Reviewer #1: Yes

Reviewer #2: Yes

3. Have the authors made all data underlying the findings in their manuscript fully available?

Reviewer #1: Yes

Reviewer #2: Yes

4. Is the manuscript presented in an intelligible fashion and written in standard English?

Reviewer #1: Yes

Reviewer #2: Yes

5. Review Comments to the Author

Reviewer #1: The article with title “Radio Frequency Interface Quality Assessment in 4G/5G: Effects of IQ Imbalance, Phase Noise, and Nonlinearities on Error Vector Magnitude” provides a well-structured analysis of RF imperfections, particularly IQ imbalance, phase noise, and amplifier nonlinearity, using Simulink models. The methodology is well-documented, and the results effectively support the conclusions. However, additional validation with real-world experimental measurements would strengthen the findings.

On the one hand, the manuscript is generally well-written, but some sentences could be restructured for better readability. A minor language revision would improve clarity. It is especially important for authors to check all spelling hyphens that appear in the text. Many of them are unnecessary, or are simply incorrect.

On the other hand, some figures, such as those showing spectral characteristics, could benefit from higher resolution and clearer labels for better visualization.

Finally, it would be remarkable if the authors consider validating the Simulink results with empirical hardware measurements.

The discussion section effectively connects the results to real-world applications, but it would be helpful to explicitly mention any limitations of the study and potential areas for future research.

Overall, the manuscript presents valuable findings relevant to 4G/5G RF interface quality assessment and is a strong contribution to the field.

Finally, it should be noted that the article shows no signs of dual publication and meets all the requirements of publication ethics and research ethics.

Reviewer #2: The manuscript presents the effect of RF interface imperfections in the communication quality assessment in 4G/5G. Several parameters of RF interface are researched for its significance in the communication and the results are analyzed using Simulink. However, the reviewer require the following points to be addressed:

1. The novelty of the proposed work is difficult to understand. Most of the conclusions are from the earlier reported work.

2. Though hardware demonstration could add significant appreciation, the simulation framework need to be highlighted with major contribution from the authors.

3. The manuscript quotes chapter 3 at two different locations for which the reference should be available.

4. Please explain fig. 4 and fig. 5 clearly with all the clarifications of the arrows and symbols presented.

6. PLOS authors have the option to publish the peer review history of their article (what does this mean? ). If published, this will include your full peer review and any attached files.

**Do you want your identity to be public for this peer review?** For information about this choice, including consent withdrawal, please see our Privacy Policy .

Reviewer #1: No

Reviewer #2: No

---

## [Author Response · Author response to Decision Letter 1]

7 Apr 2025

Look at the attachesd file, please.

---

## [Decision Letter · Decision Letter 1]

22 Apr 2025

Radio Frequency Interface Quality Assessment in 4G/5G: Effects of IQ Imbalance, Phase Noise, and Nonlinearities on Error Vector Magnitude

PONE-D-25-06441R1

Dear Dr. Kovtun,

We’re pleased to inform you that your manuscript has been judged scientifically suitable for publication and will be formally accepted for publication once it meets all outstanding technical requirements.

Kind regards,

Sushank Chaudhary, Ph.D

Academic Editor

PLOS ONE

Additional Editor Comments (optional):

Reviewers' comments:

Reviewer's Responses to Questions

**Comments to the Author**

1. If the authors have adequately addressed your comments raised in a previous round of review and you feel that this manuscript is now acceptable for publication, you may indicate that here to bypass the “Comments to the Author” section, enter your conflict of interest statement in the “Confidential to Editor” section, and submit your "Accept" recommendation.

Reviewer #1: All comments have been addressed

Reviewer #2: All comments have been addressed

2. Is the manuscript technically sound, and do the data support the conclusions?

Reviewer #1: Yes

Reviewer #2: Yes

3. Has the statistical analysis been performed appropriately and rigorously? 

Reviewer #1: Yes

Reviewer #2: Yes

4. Have the authors made all data underlying the findings in their manuscript fully available?

Reviewer #1: Yes

Reviewer #2: Yes

5. Is the manuscript presented in an intelligible fashion and written in standard English?

Reviewer #1: Yes

Reviewer #2: Yes

6. Review Comments to the Author

Reviewer #1: The authors have added all the comments proposed by the reviewers and corrected all the errors detected in the text.

In my humble opinion, the manuscript is a scientific research with data that supports the conclusions with appropriate controls, replication, and sample sizes.

The paper has enough quality to be accepted in present form.

Reviewer #2: All the reviewer comments have been addressed well by the authors. There is no conflict of interest and the manuscript can be accepted.

7. PLOS authors have the option to publish the peer review history of their article (what does this mean? ). If published, this will include your full peer review and any attached files.

**Do you want your identity to be public for this peer review?** For information about this choice, including consent withdrawal, please see our Privacy Policy .

Reviewer #1: No

Reviewer #2: No

---

## [Editor Report · Acceptance letter]

PONE-D-25-06441R1

PLOS ONE

Dear Dr. Kovtun,

I'm pleased to inform you that your manuscript has been deemed suitable for publication in PLOS ONE. Congratulations! Your manuscript is now being handed over to our production team.

Kind regards,

on behalf of

Prof. Sushank Chaudhary

Academic Editor

PLOS ONE